# Some new applications of the fractional integral and four-parameter Mittag-Leffler function

**Ahmad A. Abubaker[1], Khaled Matarneh[1], Suha B. Al-Shaikh[1]\*, Mohammad Faisal Khan[2]**

**1** Faculty of Computer Studies, Arab Open University, Riyadh, Saudi Arabia, **2** Department of Basic sciences, College of Science and Theoretical Studies, Saudi Electronic University, Riyadh, Saudi Arabia

\* s.alshaikh@arabou.edu.sa

**Data Availability Statement:** Data are contained within the article.

**Funding:** The work was funded through the Arab Open University research fund No. (AOUKSA-524008). This research receive no external funding.

## Abstract

The article reveals new applications of the four-parameter Mittag-Leffler function (MLF) in geometric function theory (GFT), using fractional calculus notions. The purpose of this study is to propose and explore a new integral operator of order $\lambda$ using fractional calculus and the four-parameter MLF. The techniques of differential subordination theory are employed in order to derive certain univalence conditions for the newly defined fractional calculus operator involving the Mittag-Leffler function. In the proved theorems and corollaries of the paper, it is specified that the fractional integral operator of the four parameter MLF satisfies the conditions to be starlike and convex. It is also proved that the newly defined operator is a starlike, convex, and close-to-convex function of positive and negative orders, respectively. The geometric properties demonstrated for the fractional integral of the four-parameter MLF show that this function could be a valuable resource for developing the study of geometric functions theory, differential subordination, and superordination theory.

## 1. Introduction

Recently, fractional calculus (FC) has become a significant field of mathematical analysis, both in terms of theoretical research and practical applications. It has become an indispensable instrument for modeling and analysis, and has had a considerable influence on a wide range of fields of study. The subject has been the focus of recent extensive reviews [1, 2], which explore its evolution and enumerate several scientific and technical areas in which it has been used.

Further investigation related to FC and univalent functions has been inspired and encouraged as a result of the current review research conducted by Srivastava [3], which emphasized the benefits of introducing FC into Geometric Function Theory (GFT). Further research has been motivated and energized by this. Miller and Mocanu [4, 5] introduced the theory of differential subordination in 1978 and 1981. This theory has attracted significant interest due to its potential to simplify the derivation of existing results and produce noteworthy outcomes when applied in studies related to analytic functions. The study of differential subordination theory has advanced significantly with the incorporation of new types of operators into the

**Competing interests:** The authors have declared
that no competing interests exist.

research. According to the recent investigation [6], integral operators are essential tools in conducting such investigations. The fractional derivative of order λ, and Riemann-Liouville fractional integral of order λ, traditional fractional calculus operators, were utilized in research on analytic functions in [7]. In [8], the generalized hypergeometric function was connected to the Riemann-Liouville fractional integral, which was previously examined in [7]. In [9], the general family of FI operators was combined with the Gauss hypergeometric function. A list of papers that may be read to track the subject's development was included in [10], which also presented a unified technique on special functions and FC operators. Recent advancements in this field have involved combining the Riemann-Liouville fractional integral with various mathematical functions, such as the Gauss hypergeometric function [11], the Confluent hypergeometric function [12], Mittag-Leffler-Confluent hypergeometric functions [13], the Libera integral operator [14], and the $q$-hypergeometric function [15].

This research builds upon existing studies in geometric function theory by introducing novel fractional operators. While the methodology draws from previous works, the incorporation of fractional calculus aspects yields new results. This study employs differential subordination methods, admissible functions theory [16], and a seminal work on geometric theory of analytic functions [17] to achieve these new findings. Notably, this combined approach has not been previously applied to research on fractional integral operators.

The novelty of this research is further enhanced by combining the Riemann-Liouville fractional integral of order λ with the form presented in [7] and the four-parameter Mittag-Leffler function. This synergy yields a new fractional integral operator that has not been explored previously. Notably, the Riemann-Liouville fractional integral has been extensively utilized in theoretical studies on new fractional operators involving hypergeometric functions, particularly in geometric function theory applications. This study pioneers the application of this versatile tool to the four-parameter Mittag-Leffler function, building upon foundational research on its geometric properties, as established in [18–21].

A four-parameter Mittag-Leffler function $E_{\alpha,\beta}^{\rho,p}(\tau)$ was presented by Shukla and Prajapati [22] in 2007. It is defined as follows:

$$E_{\alpha,\beta}^{\rho,p}(\tau) = \sum_{k=0}^{\infty} \frac{(\rho)_{pk}\tau^k}{\Gamma(k\alpha + \beta)(k)!}, \tag{1}$$

where

$$\alpha, \beta, \rho \in \mathbb{C}, \ \ p \in (0,1) \cup \mathbb{N}, \ \ Re(\alpha) > 0, \ \ Re(\rho) > 0, \ \ Re(\beta) > 0$$

and

$$(\rho)_{p(k)} = \frac{\Gamma(\rho + pk)}{\Gamma(\rho)}.$$

In general, special cases of four-parameter Mittag-Leffler function $E_{\alpha,\beta}^{\rho,p}(\tau)$ are $e^\tau$, $E_\alpha(\tau)$, $E_{\alpha,\beta}(\tau)$ and $E_{\alpha,\beta}^{\rho}(\tau)$, where $E_\alpha(\tau)$ is one-parameter Mittag-Leffler function [23], $E_{\alpha,\beta}(\tau)$ is called Wiman function and $E_{\alpha,\beta}^{\rho}(\tau)$ is Prabhakar function.

Recently, fractional calculus has been successfully used in modeling physical abnormal phenomena. As a result, generalized MLF have become frequently encountered in physical and mathematical problems, as they naturally arise when solving differential equations and FI. Consequently, mathematicians have directed more focus towards the behavior of the MLF and expanded their findings to include the complex region. See the following articles [24–28] in which researchers studied the applications of special functions. The Mittag-Leffler functions

were first described by Magnus Gösta Mittag-Leffler in 1903. Mittag-Leffler defined the classical Mittag-Leffler function $E_\alpha(\tau)$ as:

$$E_\alpha(\tau) = \sum_{k=0}^\infty \frac{\tau^k}{\Gamma(k\alpha + 1)}, \quad \alpha \in \mathbb{C}, \quad Re(\alpha) > 0.$$

Further, two-parameter Mittag-Leffler functions $E_{\alpha,\beta}(\tau)$ are defined by Wiman [29] as follows:

$$E_{\alpha,\beta}(\tau) = \sum_{k=0}^\infty \frac{\tau^k}{\Gamma(k\alpha + \beta)}, \quad \alpha, \beta \in \mathbb{C}, \quad Re(\alpha) > 0,$$

where $E_{\alpha,\beta}(\tau)$ is also called the Wiman function.

The extensive research concentrated on the fundamental characteristics of MLF as a entire functions and remained within the field of pure mathematics. After a span of thirty years, the application process for the MLF has finally started. In 1930, Hille and Tamarkin [30] used MLF to address the Abel integral equations. In 1947, Gross [31] used Mathematical Laplace Transforms to investigate the behavior of relaxation functions and creep. Barrett [32], was the first to propose a solution for a fractional differential equation using the method of Laplace transforms. In 1971, Caputo and Mainardi [33] conducted a study on fractional viscoelasticity using the MLF. In 1971, Prabhakar [34] proposed a three-parameter $E_{\alpha,\beta}^\rho(\tau)$ as an extension of MLF, based on an extensive study defined as follows:

$$E_{\alpha,\beta}^\rho(\tau) = \sum_{k=0}^\infty \frac{(\rho)_k \tau^k}{\Gamma(k\alpha + \beta)(k)!}, \quad \alpha, \beta, \rho \in \mathbb{C}, \quad Re(\alpha) > 0, \tag{2}$$

where

$$(\rho)_k = \frac{\Gamma(\rho + k)}{\Gamma(\rho)}.$$

Recently, the use of MLF, in conjunction with FC, has been used to explain the evolution of systems with memory, (see [35–40]). Motivated by the fascinating findings that have just been published [18, 41, 42] about the geometric characteristics of integral operators defined in relation to the Bessel function and the Mittag-Leffler functions. In this study, we will conduct further research on the four-parameter Mittag-Leffler function using the fractional integral operator.

This study makes significant contributions to both theoretical and practical aspects of mathematical research. By introducing new applications of the four-parameter Mittag-Leffler function in geometric function theory and developing a new integral operator using fractional calculus, this research expands the scope of geometric function theory and advances fractional calculus notions. The findings have far-reaching implications for mathematical modeling in physics, engineering, and economics, as well as signal processing and control theory applications. Additionally, the study's contributions to fractional calculus can be applied to solving fractional differential equations in various fields, including mechanics, electromagnetism, and biology. Notably, this research highlights the interactions and cross-disciplinary links between geometric function theory, fractional calculus, and differential subordination theory.

## 2. Preliminaries

After revealing the purpose of the study and providing some background information, we now present the fundamental ideas and symbols associated with geometric function theory.

Let $H(U)$ represent the general class of holomorphic functions in the unit disc $U = \{\tau \in \mathbb{C} : |\tau| < 1\}$. The notations $\overline{U} = \{\tau \in \mathbb{C} : |\tau| \leq 1\}$ and $\partial U = \{\tau \in \mathbb{C} : |\tau| = 1\}$ are connected with $U$. The class $H[a, n]$ is the subclass of $H(U)$ which is defined as:

$$H[a, n] = \{f \in H(U) : f(\tau) = a + a_n \tau^n + a_{n+1} \tau^{n+1} + \ldots, \quad \tau \in U\}.$$

Additionally, $\mathcal{A}_n \subset H(U)$ and is defined as follows:

$$\mathcal{A}_n = \{f \in H(U) : f(\tau) = \tau + a_{n+1} \tau^{n+1} + \ldots, \quad \tau \in U\}.$$

For $n = 1$, we have $\mathcal{A}_1 = \mathcal{A}$, where $\mathcal{A}$ is the class of analytic functions in $U$ that satisfy the normalized conditions $f(0) = 0$ and $f'(0) = 1$. Additionally, the research needs the following well-known classes:

(i): The class of univalent functions:

$$\mathcal{S} = \left\{ f \in \mathcal{A} : \ f \text{ is univalent}, f(0) = 0, \text{ and } f'(0) = 1 \right\}.$$

(ii): The class of starlike functions of order $\eta$:

$$\mathcal{S}^*(\eta) = \left\{ f \in \mathcal{A}: \ Re\left(\frac{\tau f'(\tau)}{f(\tau)}\right) > \eta, \quad \text{for } \eta < 1 \right\}.$$

(iii): The class of convex functions of order $\eta$:

$$\mathcal{C}(\eta) = \left\{ f \in \mathcal{A}: \ Re\left(1 + \frac{\tau f''(\tau)}{f'(\tau)}\right) > \eta, \quad \text{for } \eta < 1 \right\}.$$

(iv): The class of Carathėodory functions:

$$\mathcal{P} = \{p \in \mathcal{A}: \ p(0) = 1, \ Re(p(\tau)) > 0, \ \tau \in U\}.$$

(v): The class of close-to-convex functions:

$$\mathcal{K} = \left\{ f \in \mathcal{A}: \ \exists \varphi \in \mathcal{C}, \ Re\left(\frac{f'(\tau)}{\varphi'(\tau)}\right) > 0, \ \tau \in U \right\}. \tag{3}$$

**Remark 1** *If $\varphi(\tau) = \tau$, then (3) becomes $Re(f'(\tau)) > 0, \tau \in U$. Therefore $f$ is called close-to-convex with respect $\tau$ and it is also univalent in $U$.*

The remark given below plays an important role in proof of the main results of this article.

**Remark 2** *In ([17], Theorem. 4.4.4, p. 76), Mocanu et al. proved that $\mathcal{S}^*(\eta) \subset \mathcal{S}^*$ and $\mathcal{C}(\eta) \subset \mathcal{C}$, for $0 \leq \eta < 1$. This means that $f \in \mathcal{S}^*(\eta)$ and $f \in \mathcal{C}(\eta)$ are univalent functions. When $\eta < 0$, the functions $f$ belonging to the set $\mathcal{S}^*(\eta)$ are referred to as starlike functions of negative order, while the functions $f$ belonging to the set $\mathcal{C}(\eta)$ are known as convex functions of negative order. It is important to note that these kinds of functions may not necessarily be univalent.*

**Definition 1** *[7, 43]. Let $f_1, f_2 \in H(U)$ and $f_1$ is called subordinate to $f_2$, denoted as $f_1(\tau) \prec f_2(\tau)$, if there exists a analytic function $w$ in $U$ with $w(0) = 0$ and $|w(\tau)| < 1, \tau \in U$,*

*such that $f_1(\tau) = f_2(w(\tau))$. If $w$ is univalent, then $f_1 \prec f_2$, if and only if, $f_1(0) = f_2(0)$ and $f_1(U) \subset f_2(U)$.*

**Definition 2** [7, 8]. *The fractional integral of order $\lambda(\lambda > 0)$ is defined for a function $f$ by the following expression*:

$$D_\tau^{-\lambda}f(\tau) = \frac{1}{\Gamma(\tau)} \int\limits_0^\tau \frac{f(t)}{(\tau - t)^{1-\lambda}}\, dt. \tag{4}$$

**Remark 3** *The function $f$ is an analytic function in a simply-connected region of the $\tau$-plane containing the origin and the multiplicity of $(\tau - t)^{1-\lambda}$ is removed by requiring $\log(\tau - t)$ to be real when $(\tau - t) > 0$.*

We will use the following lemmas to prove our key findings:

**Lemma 1** ([17], *p. 84). A necessary and sufficient condition for $f \in H(U), f'(\tau) \neq 0$ to be close-to-convex if*

$$\int\limits_{\theta_1}^{\theta_2} Re\left(1 + \frac{\tau f''(\tau)}{f'(\tau)}\right) d\theta > -\pi,$$

*whenever $0 \le \theta_1 < \theta_2 < 2\pi$.*

**Lemma 2** ([17], *p. 49). Let $p \in \mathcal{P}$, and $Re(p(0)) > 0$, if*

$$Re\left(p(\tau) + \delta\frac{\tau p'(\tau)}{p(\tau)}\right) > 0, \quad \delta \in \mathbb{R}, \quad \tau \in U,$$

*then*

$$Re(p(\tau)) > 0, \quad \tau \in U.$$

## 3. Main results

The following section presents the primary results of this work, beginning with the development of the novel fractional integral operator, as described by Definition 2 and the four-parameter MLF given in (2). This study presents the first finding that provide the necessary and sufficient conditions for the novel operator to be starlike in both negative and positive orders. Following that, the same conditions develop in which the recently established operator is convex of both negative and positive orders. Moreover, it is shown that the convexity of the fractional integral of the four-parameter Mittag-Leffler function, including when $\lambda = 1$ and $\beta = 1$, shows its starlikeness. The necessary requirements for the fractional integral of the four-parameter MLF to be a close-to-convex function are derived for $\lambda = 1$ and $\beta = 1$.

Using the four-parameter MLF and the concept of fractional integrals of order $\lambda$ ($\lambda > 0$), we define and study a new fractional integral operator of the four-parameter Mittag-Leffler function as follows:

**Definition 3** *Let $\alpha, \beta, \rho \in \mathbb{C}, p \in (0,1) \cup \mathbb{N}$, where $Re(\alpha) > 0$, $Re(\rho) > 0$, $Re(\beta) > 0$ and $(\rho)_{p(k)} = \frac{\Gamma(\rho + pk)}{\Gamma(\rho)}$. The fractional integral of the four-parameter MLF are defined as:*

$$
\begin{aligned}
D_\tau^{-\lambda} E_{\alpha,\beta}^{\rho,p}(\tau) &= \frac{1}{\Gamma(\lambda)} \int_0^\tau \frac{E_{\alpha,\beta}^{\rho,p}(\tau)}{(\tau - t)^{1-\lambda}}\, dt \\
&= \frac{1}{\Gamma(\lambda)} \int_0^\tau \left( \sum_{k=0}^\infty \frac{(\rho)_{pk} t^k}{\Gamma(k\alpha + \beta)(k)!} \frac{1}{(\tau - t)^{1-\lambda}} \right) dt \\
&= \frac{1}{\Gamma(\lambda)} \left( \sum_{k=0}^\infty \frac{(\rho)_{pk}}{\Gamma(k\alpha + \beta)(k)!} \int_0^\tau \frac{t^k}{(\tau - t)^{1-\lambda}}\, dt \right) \\
&= \frac{1}{\Gamma(\lambda)} \left( \sum_{k=0}^\infty \frac{(\rho)_{pk}}{\Gamma(k\alpha + \beta)(k)!} \int_0^\tau \frac{t^k}{(\tau - t)^{1-\lambda}}\, dt \right) \\
&= \sum_{k=0}^\infty \frac{(\rho)_{pk}}{\Gamma(k\alpha + \beta)(k)!} \frac{\Gamma(k+1)}{\Gamma(k+\lambda+1)} \tau^{k+\lambda}.
\end{aligned}
\tag{5}
$$

**Remark 4** *The following are the significant characteristics of the fractional integral (FI) operator of the four-parameter Mittag-Leffler function:*

*(i): The series expansion of $D_\tau^{-\lambda} E_{\alpha,\beta}^{\rho,p}(\tau)$ is:*

$$
D_\tau^{-\lambda} E_{\alpha,\beta}^{\rho,p}(\tau) = \frac{1}{\Gamma(\beta)\Gamma(\lambda+1)} \tau^\lambda + \frac{(\rho)_p}{\Gamma(\alpha+\beta)\Gamma(\lambda+2)} \tau^{\lambda+1} + \dots.
\tag{6}
$$

*(ii): The derivative of (6) w.r.t.$\tau$ is:*

$$
(D_\tau^{-\lambda} E_{\alpha,\beta}^{\rho,p}(\tau))' = \frac{\lambda}{\Gamma(\beta)\Gamma(\lambda+1)} \tau^{\lambda-1} + \frac{(\lambda+1)(\rho)_p}{\Gamma(\alpha+\beta)\Gamma(\lambda+2)} \tau^\lambda + \dots,
$$

*then we see that $D_\tau^{-\lambda} E_{\alpha,\beta}^{\rho,p}(\tau) \notin \mathcal{A}$.*

*(iii): For $\lambda = 1$ and $\beta = 1$, then*

$$
\begin{aligned}
D_\tau^{-1} E_{\alpha,1}^{\rho,p}(\tau) &= \sum_{k=0}^\infty \frac{(\rho)_{pk}}{\Gamma(k\alpha + \beta)(k)!} \frac{\Gamma(k+1)}{\Gamma(k+\lambda+1)} \tau^{k+1} \\
&= \tau + \frac{(\rho)_p}{\Gamma(\alpha+1)\Gamma(3)} \tau^2 + \dots,
\end{aligned}
\tag{7}
$$

*(iv): The derivative of $D_\tau^{-1} E_{\alpha,1}^{\rho,p}(\tau)$ is*

$$
(D_\tau^{-1} E_{\alpha,1}^{\rho,p}(\tau))' = 1 + \frac{2(\rho)_p}{\Gamma(\alpha+1)\Gamma(3)} \tau + \dots
$$

*and $(D_\tau^{-1} E_{\alpha,1}^{\rho,p}(0))' = 1 \neq 0$, which shows that $D_\tau^{-1} E_{\alpha,1}^{\rho,p}(\tau) \in \mathcal{A}$.*

**Note that:** The operator $D_\tau^{-1} E_{\alpha,1}^{\rho,p}(\tau)$ will also play an essential role in the investigation of the classes of starlike, convex, and close-to-convex functions.

The theorem shown below provide the necessary and sufficient condition for the operator $D_\tau^{-\lambda} E_{\alpha,\beta}^{\rho,p}(\tau)$ of the four-parameter Mittag-Leffler function, as described in (5), to be starlike of negative and positive order.

**Theorem 1** *Let $D_\tau^{-\lambda} E_{\alpha,\beta}^{\rho,p}(\tau)$ be given by* (5) *and $h(\tau) = \frac{1+\tau}{1-\tau} \in C$. If*

$$\frac{\tau\left(D_\tau^{-\lambda} E_{\alpha,\beta}^{\rho,p}(\tau)\right)'}{D_\tau^{-\lambda} E_{\alpha,\beta}^{\rho,p}(\tau)} \prec \frac{1+\tau}{1-\tau}, \quad \tau \in U. \tag{8}$$

*Then*

$$Re\left(\frac{\tau\left(D_\tau^{-\lambda} E_{\alpha,\beta}^{\rho,p}(\tau)\right)'}{D_\tau^{-\lambda} E_{\alpha,\beta}^{\rho,p}(\tau)}\right) > 0, \; \tau \in U.$$

**Proof:** Since $h(\tau) \in C$, $h(0) = 1$ and

$$h(U) = \{\tau \in \mathbb{C} : Re\tau > 0\}$$

is a convex domain. Hence (8) becomes

$$Re\left(\frac{\tau\left(D_\tau^{-\lambda} E_{\alpha,\beta}^{\rho,p}(\tau)\right)'}{D_\tau^{-\lambda} E_{\alpha,\beta}^{\rho,p}(\tau)}\right) > Re(h(\tau)) > 0, \quad \tau \in U.$$

Therefore, we have

$$Re\left(\frac{\tau\left(D_\tau^{-\lambda} E_{\alpha,\beta}^{\rho,p}(\tau)\right)'}{D_\tau^{-\lambda} E_{\alpha,\beta}^{\rho,p}(\tau)}\right) > 0, \quad \tau \in U. \tag{9}$$

Thus, the theorem is proved.

**Remark 5** *Since from* (6)*, we see that $\left(D_\tau^{-\lambda} E_{\alpha,\beta}^{\rho,p}\right)'(0) = 0$, which shows that $D_\tau^{-\lambda} E_{\alpha,\beta}^{\rho,p}(\tau) \notin \mathcal{A}$, and the inequality* (9) *shows that $D_\tau^{-\lambda} E_{\alpha,\beta}^{\rho,p}(\tau) \notin \mathcal{S}^*$.*

**Remark 6** *When $\lambda = 1$ and $\beta = 1$, and considering* (7)*, we have*

$$D_\tau^{-1} E_{\alpha,1}^{\rho,p}(0) = 0$$

*and*

$$(D_\tau^{-1} E_{\alpha,1}^{\rho,p})'(0) = 1.$$

*Therefore, $D_\tau^{-1} E_{\alpha,1}^{\rho,p}(\tau) \in \mathcal{A}$.*

The following corollary for $D_\tau^{-1} E_{\alpha,1}^{\rho,p}(\tau)$ is as follows:

**Corollary 1** *Let $D_\tau^{-1} E_{\alpha,1}^{\rho,p}(\tau)$ be given in* (7) *and consider $h(\tau) = \frac{1+\tau}{1-\tau} \in C$. If*

$$\frac{\tau(D_\tau^{-1} E_{\alpha,1}^{\rho,p}(\tau))'}{D_\tau^{-1} E_{\alpha,1}^{\rho,p}(\tau)} \prec \frac{1+\tau}{1-\tau}, \quad \tau \in U,$$

*then*

$$Re\left(\frac{\tau(D_\tau^{-1}E_{\alpha,1}^{\rho,p}(\tau))'}{D_\tau^{-1}E_{\alpha,1}^{\rho,p}(\tau)}\right) > 0, \ \tau \in U. \tag{10}$$

*Implies that* $D_\tau^{-1}E_{\alpha,1}^{\rho,p}(\tau) \in \mathcal{S}^*$.

**Proof:** Since from (7), we see that $(D_\tau^{-1}E_{\alpha,1}^{\rho,p})'(0) = 1$ and from (10), we have $D_\tau^{-1}E_{\alpha,1}^{\rho,p}(\tau) \in \mathcal{S}^*$ and $D_\tau^{-1}E_{\alpha,1}^{\rho,p}(\tau) \in \mathcal{S}$.

**Example 1** *Let* $q(\tau) = \frac{1+\tau}{1-\tau}$ *with* $q(0) = 1$, $q'(\tau) = \frac{2}{(1-z)^2}$ *and* $q''(\tau) = \frac{4}{(1-z)^3}$. *Now we see*

$$Re\left(1 + \frac{\tau q''(\tau)}{q'(\tau)}\right) = Re\left(\frac{1+\tau}{1-\tau}\right)$$

$$= Re\left(\frac{1 + r\cos\theta + ir\sin\theta}{1 - r\cos\theta - ir\sin\theta}\right).$$

*After some simple calculations, we have*

$$Re\left(1 + \frac{\tau q''(\tau)}{q'(\tau)}\right) = Re\left(\frac{1 - r^2 + 2ir\sin\theta}{1 + r^2 - 2r\cos\theta}\right).$$

*The equating the real part of*

$$Re\left(1 + \frac{\tau q''(\tau)}{q'(\tau)}\right) = \frac{1 - r^2}{1 + r^2 - 2r\cos\theta} > 0,$$

*which shows that* $q(\tau) = \frac{1+\tau}{1-\tau}$ *is convex. For* $\lambda = 1$ *and* $\beta = 1$, *we have*

$$D_\tau^{-1}E_{\alpha,1}^{\rho,p}(\tau) = \tau + \frac{(\rho)_p}{\Gamma(\alpha+1)\Gamma(3)}\tau^2$$

*and*

$$\frac{\tau(D_\tau^{-1}E_{\alpha,1}^{\rho,p}(\tau))'}{D_\tau^{-1}E_{\alpha,1}^{\rho,p}(\tau)} = 1 + \frac{(\rho)_p}{2\Gamma(\alpha+1)}\tau.$$

*From* (10), *we have*

$$1 + \frac{(\rho)_p}{2\Gamma(\alpha+1)}\tau \prec q(\tau).$$

*Induces that*

$$Re\left(\frac{\tau(D_\tau^{-1}E_{\alpha,1}^{\rho,p}(\tau))'}{D_\tau^{-1}E_{\alpha,1}^{\rho,p}(\tau)}\right) = Re\left(1 + \frac{(\rho)_p}{2\Gamma(\alpha+1)}\right) > 0.$$

**Remark 7** *Here we show that* $\frac{\tau\left(D_\tau^{-1}E_{\alpha,1}^{\rho,p}(\tau)\right)'}{D_\tau^{-1}E_{\alpha,1}^{\rho,p}(\tau)} \in \mathcal{P}$.

**Proof:** Let $g(\tau) = \frac{\tau\left(D_\tau^{-1}E_{\alpha,1}^{\rho,p}(\tau)\right)'}{D_\tau^{-1}E_{\alpha,1}^{\rho,p}(\tau)}$, such that $g(0) = 1$ and from (10), we have $Re(g(\tau)) > 0$,

$\tau \in U$, implies that $g(\tau) \in \mathcal{P}$. Hence $\frac{\tau\left(D_\tau^{-1}E_{\alpha,1}^{\rho,p}(\tau)\right)'}{D_\tau^{-1}E_{\alpha,1}^{\rho,p}(\tau)} \in \mathcal{P}$.

In the next result, we determine the necessary and sufficient condition for the Fractional integral operator $\left( D_\tau^{-\lambda} E_{\alpha,\beta}^{\rho,p}(\tau) \right)$ of the four parameters Mittag-Leffler function introduced in (5) is starlike of positive order.

**Theorem 2** *Let $D_\tau^{-\lambda} E_{\alpha,\beta}^{\rho,p}(\tau)$ be given by (5) and consider $h(\tau) = \frac{\tau}{1+\tau} \in \mathcal{C}$. If*

$$\frac{\tau \left( D_\tau^{-\lambda} E_{\alpha,\beta}^{\rho,p}(\tau) \right)'}{D_\tau^{-\lambda} E_{\alpha,\beta}^{\rho,p}(\tau)} \prec \frac{\tau}{1+\tau}, \quad \tau \in U, \tag{11}$$

*then*

$$Re \left( \frac{\tau \left( D_\tau^{-\lambda} E_{\alpha,\beta}^{\rho,p}(\tau) \right)'}{D_\tau^{-\lambda} E_{\alpha,\beta}^{\rho,p}(\tau)} \right) > \frac{1}{2}, \tau \in U. \tag{12}$$

**Proof:** First of all, we show that $h(\tau) = \frac{\tau}{1+\tau} \in \mathcal{C}$. Let

$$h'(\tau) = \frac{1}{(1+\tau)^2} \text{ and } h''(\tau) = \frac{-2}{(1+\tau)^3}.$$

Thus, we have

$$Re \left( 1 + \frac{\tau h''(\tau)}{h'(\tau)} \right) = Re \left( \frac{1-\tau}{1+\tau} \right) > 0,$$

implies

$$Re \left( 1 + \frac{\tau h''(\tau)}{h'(\tau)} \right) > 0. \tag{13}$$

where $h(0) = 0$ and $h'(0) = 1$. Thus from (13), we have $h(\tau) \in \mathcal{C}$. It follows that $h(U) = \{\tau \in \mathbb{C} : Re\tau > \frac{1}{2}\}$ is convex domain and (11) is equivalent to:

$$Re \left( \frac{\tau \left( D_\tau^{-\lambda} E_{\alpha,\beta}^{\rho,p}(\tau) \right)'}{D_\tau^{-\lambda} E_{\alpha,\beta}^{\rho,p}(\tau)} \right) > Re \frac{\tau}{1+\tau} > \frac{1}{2}, \quad \tau \in U,$$

implies that

$$Re \left( \frac{\tau \left( D_\tau^{-\lambda} E_{\alpha,\beta}^{\rho,p}(\tau) \right)'}{D_\tau^{-\lambda} E_{\alpha,\beta}^{\rho,p}(\tau)} \right) > \frac{1}{2}, \quad \tau \in U. \tag{14}$$

Thus, theorem is proved.

**Remark 8** *Since from (6), we see that $\left( D_\tau^{-\lambda} E_{\alpha,\beta}^{\rho,p} \right)'(0) = 0$, which show that relation (14) does not imply that $D_\tau^{-\lambda} E_{\alpha,\beta}^{\rho,p}(\tau) \in \mathcal{S}^* \left( \frac{1}{2} \right)$, hence $D_\tau^{-\lambda} E_{\alpha,\beta}^{\rho,p}(\tau) \notin \mathcal{S}^* \left( \frac{1}{2} \right)$.*

**Corollary 2** *Let $D_\tau^{-1} E_{\alpha,1}^{\rho,p}(\tau)$ be given by (7) and consider $h(\tau) = \frac{\tau}{1+\tau} \in \mathcal{C}$. If*

$$\frac{\tau (D_\tau^{-1} E_{\alpha,1}^{\rho,p}(\tau))'}{D_\tau^{-1} E_{\alpha,1}^{\rho,p}(\tau)} \prec \frac{\tau}{1+\tau}, \quad \tau \in U.$$

*Then*

$$Re\left(\frac{\tau(D_\tau^{-1}E_{\alpha,1}^{\rho,p}(\tau))'}{D_\tau^{-1}E_{\alpha,1}^{\rho,p}(\tau)}\right) > \frac{1}{2}, \ \tau \in U. \tag{15}$$

*Implies that* $D_\tau^{-1}E_{\alpha,1}^{\rho,p}(\tau) \in \mathcal{S}^*\left(\frac{1}{2}\right)$. *Also implies that* $D_\tau^{-1}E_{\alpha,1}^{\rho,p}(\tau) \in \mathcal{S}^*$.

**Proof:** Since from (7), we see that $(D_\tau^{-1}E_{\alpha,1}^{\rho,p})'(0) = 1$, and from (15) we have $D_\tau^{-1}E_{\alpha,1}^{\rho,p}(\tau) \in \mathcal{S}^*\left(\frac{1}{2}\right)$. Since $\mathcal{S}^*(\eta) \subset \mathcal{S}^*$ for $0 \le \eta < 1$. Therefore $D_\tau^{-1}E_{\alpha,1}^{\rho,p}(\tau) \in \mathcal{S}$.

The following result determines the necessary and sufficient conditions for the operator $\left(D_\tau^{-\lambda}E_{\alpha,\beta}^{\rho,p}(\tau)\right)$, defined in (5), to be starlike of negative order.

**Theorem 3** *Let* $D_\tau^{-\lambda}E_{\alpha,\beta}^{\rho,p}(\tau)$ *be given in* (5) *and* $h(\tau) = \frac{\tau}{1-\tau} \in \mathcal{C}$. *If*

$$\frac{\tau\left(D_\tau^{-\lambda}E_{\alpha,\beta}^{\rho,p}(\tau)\right)'}{D_\tau^{-\lambda}E_{\alpha,\beta}^{\rho,p}(\tau)} \prec \frac{\tau}{1-\tau}, \ \tau \in U, \tag{16}$$

*then*

$$Re\left(\frac{\tau\left(D_\tau^{-\lambda}E_{\alpha,\beta}^{\rho,p}(\tau)\right)'}{D_\tau^{-\lambda}E_{\alpha,\beta}^{\rho,p}(\tau)}\right) > -\frac{1}{2}, \ \tau \in U.$$

**Proof:** First of all, we show that $h(\tau) = \frac{\tau}{1-\tau} \in \mathcal{C}$. Let

$$h'(\tau) = \frac{1}{(1-\tau)^2} \ \text{and} \ h''(\tau) = \frac{2}{(1-\tau)^3}.$$

Thus, we have

$$Re\left(1 + \frac{\tau h''(\tau)}{h'(\tau)}\right) = Re\left(\frac{1+\tau}{1-\tau}\right) > 0.$$

Implies that

$$Re\left(1 + \frac{\tau h''(\tau)}{h'(\tau)}\right) > 0 \tag{17}$$

and $h(0) = 0$ and $h'(0) = 1$. Thus, from (17), we have $h(\tau) \in \mathcal{C}$. It follows that $h(U) = \{\tau \in \mathbb{C} : Re(\tau) > \frac{-1}{2}\}$ is convex domain and (11) is equivalent to

$$Re\left(\frac{\tau\left(D_\tau^{-\lambda}E_{\alpha,\beta}^{\rho,p}(\tau)\right)'}{D_\tau^{-\lambda}E_{\alpha,\beta}^{\rho,p}(\tau)}\right) > Re\frac{\tau}{1-\tau} > \frac{-1}{2}, \ \tau \in U.$$

Implies that

$$Re\left(\frac{\tau\left(D_\tau^{-\lambda}E_{\alpha,\beta}^{\rho,p}(\tau)\right)'}{D_\tau^{-\lambda}E_{\alpha,\beta}^{\rho,p}(\tau)}\right) > \frac{-1}{2}, \ \tau \in U. \tag{18}$$

Hence, theorem is proved.

**Remark 9** *Since from* (6), *we see* $(D_\tau^{-\lambda}E_{\alpha,\beta}^{\rho,p})'(0) = 0$, *which show that relation* (18) *does not imply that* $D_\tau^{-\lambda}E_{\alpha,\beta}^{\rho,p}(\tau) \in \mathcal{S}^*\left(-\frac{1}{2}\right)$, *hence* $D_\tau^{-\lambda}E_{\alpha,\beta}^{\rho,p}(\tau) \notin \mathcal{S}^*\left(-\frac{1}{2}\right)$.

**Corollary 3** *Let* $D_\tau^{-1} E_{\alpha,1}^{\rho,p}(\tau)$ *be given by* (7) *and* $h(\tau) = \frac{\tau}{1-\tau} \in \mathcal{C}$. *If*

$$\frac{\tau \left(D_\tau^{-1} E_{\alpha,1}^{\rho,p}(\tau)\right)'}{D_\tau^{-1} E_{\alpha,1}^{\rho,p}(\tau)} \prec \frac{\tau}{1-\tau}, \quad \tau \in U.$$

*Then*

$$Re\left(\frac{\tau \left(D_\tau^{-1} E_{\alpha,1}^{\rho,p}(\tau)\right)'}{D_\tau^{-1} E_{\alpha,1}^{\rho,p}(\tau)}\right) > -\frac{1}{2}, \ \tau \in U. \tag{19}$$

*Implies that* $D_\tau^{-1} E_{\alpha,1}^{\rho,p}(\tau) \in \mathcal{S}^* \left(-\frac{1}{2}\right)$, *but* $D_\tau^{-1} E_{\alpha,1}^{\rho,p}(\tau) \notin \mathcal{S}$.

**Proof:** Since from (7), we see that $\left(D_\tau^{-1} E_{\alpha,1}^{\rho,p}\right)'(0) = 1$, and from (19), we have $D_\tau^{-1} E_{\alpha,1}^{\rho,p}(\tau) \in \mathcal{S}^* \left(-\frac{1}{2}\right)$. Since $\mathcal{S}^*(\eta) \subset \mathcal{S}$ for $0 \leq \eta < 1$. Therefore, we see that $\frac{-1}{2} < 0$, it means that $\mathcal{S}^* \left(-\frac{1}{2}\right) \not\subseteq \mathcal{S}^*$. Hence, it is possible $D_\tau^{-1} E_{\alpha,1}^{\rho,p}(\tau)$ is not univalent in $U$.

The theorem shown below prove the necessary and sufficient condition for $D_\tau^{-\lambda} E_{\alpha,\beta}^{\rho,p}(\tau)$, as described in (5), is convex of negative and positive order, respectively.

**Theorem 4** *Let* $D_\tau^{-\lambda} E_{\alpha,\beta}^{\rho,p}(\tau)$ *be given in* (5) *and* $h(\tau) = \frac{1-\tau}{1+\tau} \in \mathcal{C}$. *If*

$$1 + \frac{\tau \left(D_\tau^{-\lambda} E_{\alpha,\beta}^{\rho,p}(\tau)\right)''}{\left(D_\tau^{-\lambda} E_{\alpha,\beta}^{\rho,p}(\tau)\right)'} \prec \frac{1-\tau}{1+\tau}, \ \tau \in U. \tag{20}$$

*Then*

$$Re\left(1 + \frac{\tau \left(D_\tau^{-\lambda} E_{\alpha,\beta}^{\rho,p}(\tau)\right)''}{\left(D_\tau^{-\lambda} E_{\alpha,\beta}^{\rho,p}(\tau)\right)'}\right) > 0, \ \tau \in U.$$

**Proof:** Since $h(\tau) = \frac{1-\tau}{1+\tau} \in \mathcal{C}$, with $h(0) = 1$ and

$$h(U) = \{\tau \in \mathbb{C} : Re(\tau) > 0\}$$

is a convex domain and (20) can be written as:

$$Re\left(1 + \frac{\tau \left(D_\tau^{-\lambda} E_{\alpha,\beta}^{\rho,p}(\tau)\right)''}{\left(D_\tau^{-\lambda} E_{\alpha,\beta}^{\rho,p}(\tau)\right)'}\right) > Re(h(\tau)) > 0, \ \tau \in U. \tag{21}$$

Hence theorem is proved.

**Remark 10** *Since from* (6), *we see* $\left(D_\tau^{-\lambda} E_{\alpha,\beta}^{\rho,p}\right)'(0) = 0$. *Therefore relation* (21) *show that* $D_\tau^{-\lambda} E_{\alpha,\beta}^{\rho,p}(\tau) \notin \mathcal{C}$.

**Corollary 4** *Let* $D_\tau^{-1} E_{\alpha,1}^{\rho,p}(\tau)$ *be given by* (7) *and* $h(\tau) = \frac{1-\tau}{1+\tau} \in \mathcal{C}$. *If*

$$1 + \frac{\tau \left(D_\tau^{-\lambda} E_{\alpha,\beta}^{\rho,p}(\tau)\right)''}{\left(D_\tau^{-\lambda} E_{\alpha,\beta}^{\rho,p}(\tau)\right)'} \prec \frac{1-\tau}{1+\tau}, \ \tau \in U.$$

*Then*

$$Re\left(1+\frac{\tau\left(D_\tau^{-\lambda}E_{\alpha,\beta}^{\rho,p}(\tau)\right)''}{\left(D_\tau^{-\lambda}E_{\alpha,\beta}^{\rho,p}(\tau)\right)'}\right) > 0, \ \tau \in U. \tag{22}$$

*Implies that* $D_\tau^{-1}E_{\alpha,1}^{\rho,p}(\tau) \in \mathcal{C}.$

**Proof:** Since from (7), we see that $(D_\tau^{-1}E_{\alpha,1}^{\rho,p})'(0) = 1$ and from (22), we have $D_\tau^{-1}E_{\alpha,1}^{\rho,p}(\tau) \in \mathcal{C}$ and $D_\tau^{-1}E_{\alpha,1}^{\rho,p}(\tau) \in \mathcal{S}.$

Now we show that $D_\tau^{-1}E_{\alpha,1}^{\rho,p}(\tau) \in \mathcal{C}$ implies that $D_\tau^{-1}E_{\alpha,1}^{\rho,p}(\tau) \in \mathcal{S}^*.$

**Theorem 5** *Let* $D_\tau^{-1}E_{\alpha,1}^{\rho,p}(\tau)$ *be given in* (7)*. If* $(D_\tau^{-1}E_{\alpha,1}^{\rho,p})(0) = 0, (D_\tau^{-1}E_{\alpha,1}^{\rho,p})'(0) = 1$ *and* $D_\tau^{-1}E_{\alpha,1}^{\rho,p}(\tau) \in \mathcal{C}$, *then* $D_\tau^{-1}E_{\alpha,1}^{\rho,p}(\tau) \in \mathcal{S}^*.$

**Proof:** Let

$$g(\tau) = \frac{\tau(D_\tau^{-1}E_{\alpha,1}^{\rho,p}(\tau))'}{D_\tau^{-1}E_{\alpha,1}^{\rho,p}(\tau)}, \tau \in U. \tag{23}$$

Using (7) in (23), we see that

$$Re(g(0)) > 0 \ \text{and} \ g(0) = 1.$$

Now differentiating (23), we get

$$g(\tau) + \frac{\tau g'(\tau)}{g(\tau)} = 1 + \frac{\tau\left(D_\tau^{-\lambda}E_{\alpha,\beta}^{\rho,p}(\tau)\right)''}{\left(D_\tau^{-\lambda}E_{\alpha,\beta}^{\rho,p}(\tau)\right)'}, \ \tau \in U. \tag{24}$$

As we know that $D_\tau^{-1}E_{\alpha,1}^{\rho,p}(\tau) \in \mathcal{C}$, then

$$Re\left(1+\frac{\tau\left(D_\tau^{-\lambda}E_{\alpha,\beta}^{\rho,p}(\tau)\right)''}{\left(D_\tau^{-\lambda}E_{\alpha,\beta}^{\rho,p}(\tau)\right)'}\right) > 0, \ \tau \in U.$$

Therefore, (24), becomes that

$$Re\left(g(\tau)+\frac{\tau g'(\tau)}{g(\tau)}\right) > 0, \ \tau \in U. \tag{25}$$

Using the Lemma 2, for $\delta = 1$. Inequality (25) becomes

$$Reg(\tau) > 0, \ \tau \in U. \tag{26}$$

Using relation (23) in (26), we have

$$Re\left(\frac{\tau(D_\tau^{-1}E_{\alpha,1}^{\rho,p}(\tau))'}{D_\tau^{-1}E_{\alpha,1}^{\rho,p}(\tau)}\right) > 0, \ \tau \in U. \tag{27}$$

Implies that $D_\tau^{-1}E_{\alpha,1}^{\rho,p}(\tau) \in \mathcal{S}^*.$

The theorem shown below provide the necessary and sufficient condition for the operator $D_\tau^{-\lambda}E_{\alpha,\beta}^{\rho,p}(\tau)$ of four-parameter Mittag-Leffler function, as described in (5), to be convex for $0 \leq \eta < 1.$

**Theorem 6** *Let* $D_\tau^{-\lambda} E_{\alpha,\beta}^{\rho,p}(\tau)$ *be given by* (5) *and let* $h(\tau) = \frac{1}{1-\tau} \in \mathcal{C}$. *If*

$$1 + \frac{\tau \left( D_\tau^{-\lambda} E_{\alpha,\beta}^{\rho,p}(\tau) \right)''}{\left( D_\tau^{-\lambda} E_{\alpha,\beta}^{\rho,p}(\tau) \right)'} \prec \frac{1}{1-\tau}, \quad \tau \in U. \tag{28}$$

*Then*

$$Re \left( 1 + \frac{\tau \left( D_\tau^{-\lambda} E_{\alpha,\beta}^{\rho,p}(\tau) \right)''}{\left( D_\tau^{-\lambda} E_{\alpha,\beta}^{\rho,p}(\tau) \right)'} \right) > \frac{1}{2}, \quad \tau \in U.$$

**Proof:** First of all, we show that $h(\tau) = \frac{1}{1-\tau} \in \mathcal{C}$. Let

$$h'(\tau) = \frac{1}{(1-\tau)^2} \quad \text{and} \quad h''(\tau) = \frac{2}{(1-\tau)^3}.$$

Thus, we have

$$Re \left( 1 + \frac{\tau h''(\tau)}{h'(\tau)} \right) = Re \left( \frac{1+\tau}{1-\tau} \right) > 0,$$

implies

$$Re \left( 1 + \frac{\tau h''(\tau)}{h'(\tau)} \right) > 0 \tag{29}$$

and $h(0) = 0$ and $h'(0) = 1$ and from (29), we have $h(\tau) \in \mathcal{C}$. It follows that $h(U) = \{\tau \in \mathbb{C} : Re\tau > \frac{1}{2}\}$ is convex domain and

$$Re \left( 1 + \frac{\tau \left( D_\tau^{-\lambda} E_{\alpha,\beta}^{\rho,p}(\tau) \right)''}{\left( D_\tau^{-\lambda} E_{\alpha,\beta}^{\rho,p}(\tau) \right)'} \right) > Re \frac{1}{1-\tau} > \frac{1}{2}, \quad \tau \in U.$$

Implies that

$$Re \left( 1 + \frac{\tau \left( D_\tau^{-\lambda} E_{\alpha,\beta}^{\rho,p}(\tau) \right)''}{\left( D_\tau^{-\lambda} E_{\alpha,\beta}^{\rho,p}(\tau) \right)'} \right) > \frac{1}{2}, \quad \tau \in U. \tag{30}$$

Hence theorem is proved.

**Remark 11** *Since* $\left( D_\tau^{-\lambda} E_{\alpha,\beta}^{\rho,p} \right)'(0) = 0$, *therefore relation* (30) *show that* $D_\tau^{-\lambda} E_{\alpha,\beta}^{\rho,p}(\tau) \notin \mathcal{C}\left(\frac{1}{2}\right)$.

Considering the relation (7), we prove that $D_\tau^{-1} E_{\alpha,1}^{\rho,p}(\tau) \in \mathcal{C}$.

**Corollary 5** *Let* $D_\tau^{-1} E_{\alpha,1}^{\rho,p}(\tau)$ *be given by* (7) *and consider* $h(\tau) = \frac{1}{1-\tau} \in \mathcal{C}$. *If*

$$1 + \frac{\tau (D_\tau^{-1} E_{\alpha,1}^{\rho,p}(\tau))''}{(D_\tau^{-1} E_{\alpha,1}^{\rho,p}(\tau))'} \prec \frac{1}{1-\tau}, \quad \tau \in U.$$

*Then*

$$Re\left(1 + \frac{\tau(D_\tau^{-1} E_{\alpha,1}^{\rho,p}(\tau))''}{(D_\tau^{-1} E_{\alpha,1}^{\rho,p}(\tau))'}\right) > \frac{1}{2}, \; \tau \in U. \tag{31}$$

*Then $D_\tau^{-1} E_{\alpha,1}^{\rho,p}(\tau) \in \mathcal{C}$.*

**Proof:** Since $(D_\tau^{-1} E_{\alpha,1}^{\rho,p})'(0) = 1 \neq 0$, then inequality (31) gives that $D_\tau^{-1} E_{\alpha,1}^{\rho,p}(\tau) \in \mathcal{C}(\frac{1}{2})$. According to the Remark 3, we have $D_\tau^{-1} E_{\alpha,1}^{\rho,p}(\tau) \in \mathcal{C}$, because $\eta = \frac{1}{2} \in [0, 1)$.

The theorem shown below provide the necessary and sufficient condition for $D_\tau^{-\lambda} E_{\alpha,\beta}^{\rho,p}(\tau)$, as described in (5), is the convex of negative order.

**Theorem 7** *Let $D_\tau^{-\lambda} E_{\alpha,\beta}^{\rho,p}(\tau)$ be given in (5) and $h(\tau) = \frac{1-2\tau}{1+\tau} \in \mathcal{C}$. If*

$$1 + \frac{\tau\left(D_\tau^{-\lambda} E_{\alpha,\beta}^{\rho,p}(\tau)\right)''}{\left(D_\tau^{-\lambda} E_{\alpha,\beta}^{\rho,p}(\tau)\right)'} \prec \frac{1-2\tau}{1+\tau}, \; \tau \in U. \tag{32}$$

*Then*

$$Re\left(1 + \frac{\tau\left(D_\tau^{-\lambda} E_{\alpha,\beta}^{\rho,p}(\tau)\right)''}{\left(D_\tau^{-\lambda} E_{\alpha,\beta}^{\rho,p}(\tau)\right)'}\right) > -\frac{1}{2}, \; \tau \in U.$$

**Proof:** First of all, we show that $h(\tau) = \frac{1-2\tau}{1+\tau} \in \mathcal{C}$. Let

$$h'(\tau) = \frac{-3}{(1+\tau)^2} \; \text{ and } \; h''(\tau) = \frac{6}{(1+\tau)^3}.$$

Thus, we have

$$Re\left(1 + \frac{\tau h''(\tau)}{h'(\tau)}\right) = Re\left(\frac{1-\tau}{1+\tau}\right) > 0.$$

Implies

$$Re\left(1 + \frac{\tau h''(\tau)}{h'(\tau)}\right) > 0 \tag{33}$$

and $h(0) = 0$ and $h'(0) = 1$. Thus, from (33), we have $h(\tau) \in \mathcal{C}$. It follows that $h(U) = \{\tau \in \mathbb{C} : Re(\tau) > \frac{-1}{2}\}$ is convex domain and (32) is equivalent to:

$$Re\left(1 + \frac{\tau\left(D_\tau^{-\lambda} E_{\alpha,\beta}^{\rho,p}(\tau)\right)''}{\left(D_\tau^{-\lambda} E_{\alpha,\beta}^{\rho,p}(\tau)\right)'}\right) > Re\frac{1-2\tau}{1+\tau} > \frac{-1}{2}, \; \tau \in U.$$

Implies that

$$Re\left(1 + \frac{\tau\left(D_\tau^{-\lambda} E_{\alpha,\beta}^{\rho,p}(\tau)\right)''}{\left(D_\tau^{-\lambda} E_{\alpha,\beta}^{\rho,p}(\tau)\right)'}\right) > -\frac{1}{2}, \; \tau \in U. \tag{34}$$

Hence theorem is proved.

**Remark 12** *Since $(D_\tau^{-\lambda} E_{\alpha,\beta}^{\rho,p})'(0) = 0$, therefore relation (34) shows that $D_\tau^{-\lambda} E_{\alpha,\beta}^{\rho,p}(\tau) \notin \mathcal{C}\left(-\frac{1}{2}\right)$.*

Considering the relation (7), we prove that $D_\tau^{-1}E_{\alpha,1}^{\rho,p}(\tau) \in \mathcal{C}$.

**Corollary 6** *Let* $D_\tau^{-1}E_{\alpha,1}^{\rho,p}(\tau)$ *be given by* (7) *and* $h(\tau) = \frac{1-2\tau}{1+\tau} \in \mathcal{C}$. *If*

$$1 + \frac{\tau(D_\tau^{-1}E_{\alpha,1}^{\rho,p}(\tau))''}{(D_\tau^{-1}E_{\alpha,1}^{\rho,p}(\tau))'} \prec \frac{1-2\tau}{1+\tau}, \quad \tau \in U.$$

*Then*

$$Re\left(1 + \frac{\tau(D_\tau^{-1}E_{\alpha,1}^{\rho,p}(\tau))''}{(D_\tau^{-1}E_{\alpha,1}^{\rho,p}(\tau))'}\right) > -\frac{1}{2}, \quad \tau \in U. \tag{35}$$

*Then,* $D_\tau^{-1}E_{\alpha,1}^{\rho,p}(\tau) \in \mathcal{C}\left(-\frac{1}{2}\right)$.

**Proof:** Since $(D_\tau^{-1}E_{\alpha,1}^{\rho,p})'(0) = 1 \neq 0$, then inequality (35) gives that $D_\tau^{-1}E_{\alpha,1}^{\rho,p}(\tau) \in \mathcal{C}\left(-\frac{1}{2}\right)$. According to the Remark 2, we see that $D_\tau^{-1}E_{\alpha,1}^{\rho,p}(\tau) \notin \mathcal{C}$, because $\eta = -\frac{1}{2} \notin [0,1)$. According to remarks 2, it is not necessarily univalent.

Apply the proof of above corollary, we prove that $D_\tau^{-1}E_{\alpha,1}^{\rho,p}(\tau)$ defined in (7) is univalent and close-to-convex.

**Theorem 8** *Let* $D_\tau^{-1}E_{\alpha,1}^{\rho,p}(\tau)$ *is given in* (7) *satisfy*

$$Re\left(1 + \frac{\tau(D_\tau^{-1}E_{\alpha,1}^{\rho,p}(\tau))''}{(D_\tau^{-1}E_{\alpha,1}^{\rho,p}(\tau))'}\right) > -\frac{1}{2}, \quad , \tau \in U,$$

*then* $D_\tau^{-1}E_{\alpha,1}^{\rho,p}(\tau) \in \mathcal{K}$ *and* $D_\tau^{-1}E_{\alpha,1}^{\rho,p}(\tau) \in \mathcal{S}$.

**Proof:** Using the hypothesis, we see

$$\int_{\theta_1}^{\theta_2} Re\left(1 + \frac{\tau(D_\tau^{-1}E_{\alpha,1}^{\rho,p}(\tau))''}{(D_\tau^{-1}E_{\alpha,1}^{\rho,p}(\tau))'}\right) d\theta \quad > \quad \int_{\theta_1}^{\theta_2} -\frac{1}{2} d\theta$$

$$= \quad -\frac{1}{2}(\theta_2 - \theta_1)$$

$$> \quad -\pi.$$

Since $0 \leq \theta_1 < \theta_2 < 2\pi$. Applying the Lemma 1, we have, $D_\tau^{-1}E_{\alpha,1}^{\rho,p}(\tau) \in \mathcal{K}$ and $D_\tau^{-1}E_{\alpha,1}^{\rho,p}(\tau) \in \mathcal{S}$. Hence $D_\tau^{-1}E_{\alpha,1}^{\rho,p}(\tau)$ is a close-to-convex and also univalent.

## 4. Discussion

The primary objective of this research is to introduce a new FI operator based on the four-parameter MLF and to start research studies into its geometric characteristics using the tools of differential subordination theory. The Definition 3 of FI of the four parametric MLF given by relation (5) is introduced using the FI of order $\lambda$ defined in Definition 2 known by relation (4) and the Mittag-Leffler function shown given in (2). The introduction provides the necessary concepts for this research and presents the key tools and two lemmas used for obtaining the new findings. Section 3 contains 8 theorems, 6 corollaries, and numerous observations that provide insights into the criteria for the univalence of the FI operator $D_\tau^{-\lambda}E_{\alpha,\beta}^{\rho,p}(\tau)$ associated with the four parameters MLF. Theorems 1, 2 and 3 together with Corollaries 1, 2, and 3 provide the necessary and sufficient conditions for $D_\tau^{-\lambda}E_{\alpha,\beta}^{\rho,p}(\tau)$ is starlike of negative and positive order. Theorems 4, 6, and 7 together with Corollaries 4, 5, and 6 provide the necessary and sufficient conditions for $D_\tau^{-\lambda}E_{\alpha,\beta}^{\rho,p}(\tau)$ is convex of negative and positive order. Using the Corollary 4, in Theorem 5, then it shows that convexity of the form $D_\tau^{-1}E_{\alpha,1}^{\rho,p}(\tau)$ implies its starlikeness. In

addition, by using the information provided in Corollary 6, the last result 8 presented in this paper establishes a criterion for the operator $D_\tau^{-1} E_{\alpha,1}^{\rho,p}(\tau)$ that has the properties of being close-to-convex and univalent.

## 5. Conclusion

The findings of this work are significant for research problems connected with special functions and integral FC in GFT. The results presented here may be used in the future to create new subclasses of analytic functions with specific geometric features provided by the characteristics of $D_\tau^{-\lambda} E_{\alpha,\beta}^{\rho,p}(\tau)$ provided by (5), which have already been shown in this article.

Future research may also be conducted on the operator $D_\tau^{-\lambda} E_{\alpha,\beta}^{\rho,p}(\tau)$ described in this study, which is connected to the extensions of differential subordination theories identified as fuzzy differential subordination and strong differential subordination which have been developed in previously.

Recent studies [44–46] have utilized distinct fractional operators to explore two separate concepts, providing valuable insights for future research. To further verify the significant results obtained in this study, additional research is necessary, particularly when applying the Riemann-Liouville fractional derivative of order λ to the $D_\tau^{-\lambda} E_{\alpha,\beta}^{\rho,p}(\tau)$.

Building on the foundational work in [47–49], researchers can leverage $q$-calculus, $q$-Mittag-Leffler functions, and $q$-fractional integrals of order λ to investigate similar properties, as discussed in this article. By exploring these areas, researchers can advance our understanding of fractional calculus and its potential applications in various fields.

## Author Contributions

**Funding acquisition:** Suha B. Al-Shaikh.

**Supervision:** Suha B. Al-Shaikh.

**Writing – original draft:** Ahmad A. Abubaker.

**Writing – review & editing:** Ahmad A. Abubaker, Khaled Matarneh, Suha B. Al-Shaikh, Mohammad Faisal Khan.

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
