## [Decision Letter · Decision Letter 0]

20 Nov 2024

PONE-D-24-38839Some new applications of the fractional integral and four-parameter Mittag-Leffler functionPLOS ONE

Dear Dr. B. Al-Shaikh,

Thank you for submitting your manuscript to PLOS ONE. After careful consideration, we feel that it has merit but does not fully meet PLOS ONE’s publication criteria as it currently stands. Therefore, we invite you to submit a revised version of the manuscript that addresses the points raised during the review process. Please submit your revised manuscript by Jan 04 2025 11:59PM. If you will need more time than this to complete your revisions, please reply to this message or contact the journal office at plosone@plos.org. Please include the following items when submitting your revised manuscript:A rebuttal letter that responds to each point raised by the academic editor and reviewer(s). You should upload this letter as a separate file labeled 'Response to Reviewers'.A marked-up copy of your manuscript that highlights changes made to the original version. You should upload this as a separate file labeled 'Revised Manuscript with Track Changes'.An unmarked version of your revised paper without tracked changes. You should upload this as a separate file labeled 'Manuscript'.

We look forward to receiving your revised manuscript.

Kind regards,

Behzad Ghanbari

Academic Editor

PLOS ONE

Journal Requirements:

2. Thank you for stating the following financial disclosure: The work was funded through the Arab Open University research fund No. (AOUKSA-524008).  

3. Thank you for stating the following in the Acknowledgments Section of your manuscript: The authors extend their appreciation to the Arab Open

University for funding this work through AOU research fund no. (AOUKSA52400

Please remove any funding-related text from the manuscript and let us know how you would like to update your Funding Statement. Currently, your Funding Statement reads as follows: The work was funded through the Arab Open University research fund No. (AOUKSA-524008).

5. Comments from PLOS Editorial Office: We note that one or more reviewers has recommended that you cite specific previously published works. As always, we recommend that you please review and evaluate the requested works to determine whether they are relevant and should be cited. It is not a requirement to cite these works. We appreciate your attention to this request.

Reviewers' comments:

Reviewer's Responses to Questions

**Comments to the Author**

1. Is the manuscript technically sound, and do the data support the conclusions?

Reviewer #1: Yes

Reviewer #2: Yes

Reviewer #3: Yes

Reviewer #4: Yes

2. Has the statistical analysis been performed appropriately and rigorously? 

Reviewer #1: Yes

Reviewer #2: Yes

Reviewer #3: N/A

Reviewer #4: N/A

3. Have the authors made all data underlying the findings in their manuscript fully available?

Reviewer #1: Yes

Reviewer #2: Yes

Reviewer #3: Yes

Reviewer #4: Yes

4. Is the manuscript presented in an intelligible fashion and written in standard English?

Reviewer #1: Yes

Reviewer #2: Yes

Reviewer #3: Yes

Reviewer #4: Yes

5. Review Comments to the Author

Reviewer #1: REVIEWER’s REPORT

On the paper PONE-D-24-38839

Some new applications of the fractional integral and four-parameter Mittag-Leffler function

by Ahmad A Abubaker, Khaled Matarneh, Suha B. Al-Shaikh, Mohammad Faisal Khan

In this paper the authors establish conditions for a new integral operator of order λ defined by using fractional integral of the four-parameter Mittag-Leffler function to be starlike, convex, and close-to-convex function of positive and negative orders, respectively.

The results are new, correct and detailed. The paper is original and doesn’t contradict to ethical or policy issues, the question posed by authors is new and well defined, the methods used by authors are appropriate and well described, the data are sound and well controlled, the discussion and conclusions are well balanced, the title and abstract convey the obtained results, the writing is acceptable, the paper contains good scientific results.

The paper doesn’t require a revision.

Taking the above into consideration, I recommend the paper for publication in PLOS ONE.

1.10.2024

Reviewer #2: In this paper, the authors obtianed the theorems about the necessary and sufficient conditions for the operator of the four-parameter Mittag-Leffler function usingfractionalcalculus. Then, the authors showed the fractional integral operator of the four-parameter must satisfy the conditions of starlikeness and convexity. Finally, the study of geometric functions theory, differential subordination, and superordination theories were given. The main results are interesting. Some specific commnents are presented as follows.

1) Some definitions ands lemmas should be given as the preliminaries section.

2) The main motivations and contributions of this paper should be clarified in the introduction part.

3) Some comparsions should be given to show the advantage of this paper.

4) Do you provide some examples to show the effectiveness of main results?

Reviewer #3: In this article authors present innovative applications of the fractional integral of the four-parameter Mittag-Leffler function in the Geometric Function Theory. This study defines a new integral operator of order λ using the fractional integral and four-parameter Mittag-Leffler function. The authors employ differential subordination technique to derive geometric features satisfying univalence conditions for the newly defined fractional integral operator of Mittag-Leffler function. The paper demonstrates rigorous mathematical derivations and proofs, ensuring the correctness and reliability of the results. Using the concepts of remark 1.2, they proved that the newly defined operator is starlike, convex, and close-to-convex for positive and negative orders, respectively.

The manuscript is well-structured and clear, with precise mathematical formulations and thorough proofs. The abstract, discussion, and conclusion effectively summarize the research. However, the authors should revise the introduction section to clearly state the motivation behind this new work and explicitly address the gap between existing research and the new contributions presented. Additionally, the introduction should provide a clearer context by referencing relevant and recent articles related to their work.

1. Applications of a q-Differential Operator to a Class of Harmonic Mappings Defined by q-Mittag–Leffler Functions, Symmetry, 14, 1905 (2022), doi.org/10.3390/sym14091905.

2. Faber polynomial coefficients estimates for certain subclasses of q-Mittag-Leffler-Type analytic and bi-univalent functions, AIMS Mathematics, 7(2) (2021), 2512-2528.

3. Some Applications of Analytic Functions Associated with q-Fractional Operator, Mathematics, 11, (2023), 930.https://doi.org/10.3390/ math110409.

I would like to bring to your attention some errors and suggestions

1. In the abstract line 4: It should be “four-parameter Mittag-Leffler function ”

2. In the abstract, line 7: Replace “specified” with “proved”.

3. On page 9, line 23: Replace “specific instances “ with special cases”

4. On page 12, in the start of the line 5, remove the extra words “some new results”.

5. Check for typos error in the whole paper.

6. The conclusion section lacks clarity on future directions. Please revise and expand the conclusions to include specific future research directions

Decision: This paper by Abubaker et al. makes a substantial contribution to fractional calculus and Geometric Function Theory through differential subordination. The results are well formulated, new and easily understandable. After the above mentioned changes, the article can be published in “Plos One”.

Reviewer #4: The paper addresses a topic that provides very interesting results and is intensely investigated nowadays which concerns the embedding of fractional calculus aspects into geometric function theory. The introduction of new fractional operators is a research theme of interest at the moment and this is well evidenced in the text of the paper. The new integral operator introduced in this paper is valuable due to the properties proved in this paper which can open ways for future investigations by applying this operator in the theories of differential subordination and superordination and other fields that apply fractional operators.

The motivation of the paper is clearly presented, all the necessary notions and previously established results are highlighted in the Introduction and all the references cited in the text are relevant.

The paper is well organized and easy to follow. The proofs are correct and contain all the necessary details.

The paper needs small adjustments regarding the English language. Most important, the Abstract should be rephrased. I suggest the following:

The article reveals new applications of the four-parameter Mittag-Le­ffler function (MLF) in geometric function theory (GFT), using fractional calculus notions. The purpose of this study is to propose and explore a new integral operator of order \\lambda using fractional calculus and the four-parameter MLF. The techniques of differential subordination theory are employed in order to derive certain univalence conditions for the newly defined fractional calculus operator involving the Mittag-Le­ffler function. In the proved theorems and corollaries of the paper, it is specified that the fractional integral operator of the four-parameter MLF satisfies the conditions to be starlike and convex. It is also proved that the newly defined operator is a starlike, convex, and close-to-convex function of positive and negative orders, respectively. The geometric properties demonstrated for the fractional integral of the four-parameter MLF show that this function could be a valuable resource for developing the study of geometric functions theory, differential subordination, and superordination theories.

page 3: "Let H(U) represents" should be "Let H(U) represent" and the sentence should begin at a new paragraph.

"...is the subclass of H(U) which is define as:" should be "...is the subclass of H(U) which is defined as:"

6. PLOS authors have the option to publish the peer review history of their article (what does this mean?). If published, this will include your full peer review and any attached files.

Reviewer #1: **Yes: **Alina Alb Lupas

Reviewer #2: No

Reviewer #3: No

Reviewer #4: No

---

## [Author Response · Author response to Decision Letter 0]

19 Dec 2024

Reviewer≠1: Comments to the Author

REVIEWER’s REPORT

On the paper PONE-D-24-38839

Some new applications of the fractional integral and four-parameter Mittag-Leffler function

by Ahmad A Abubaker, Khaled Matarneh, Suha B. Al-Shaikh, Mohammad Faisal Khan

The results are new, correct and detailed. The paper is original and doesn’t contradict to ethical or policy issues, the question posed by authors is new and well defined, the methods used by authors are appropriate and well described, the data are sound and well controlled, the discussion and conclusions are well balanced, the title and abstract convey the obtained results, the writing is acceptable, the paper contains good scientific results.

The paper doesn’t require a revision.

Taking the above into consideration, I recommend the paper for publication in PLOS ONE..

Reply:

Respected Professor

Thank you for your careful reading and remarks. We have made some small changes for the improvement of our paper in the revised version. All changes are highlighted in the revised version's PDF.

Reviewer≠2: Comments to the Author

In this paper, the authors obtained the theorems about the necessary and sufficient conditions for the operator of the four-parameter Mittag-Leffler function using fractional calculus. Then, the authors showed the fractional integral operator of the four-parameter must satisfy the conditions of starlikeness and convexity. Finally, the study of geometric functions theory, differential subordination, and superordination theories were given. The main results are interesting. Some specific comments are presented as follows.

 Some definitions and lemmas should be given as the preliminaries section.

Reply to the Reviewer: Respected professor according to your suggestions we made a new sections name as preliminaries for some definitions and lemmas in the revised version.

 The main motivations and contributions of this paper should be clarified in the introduction part.

Reply to the Reviewer: we have added new paragraph in the revised version which clearly explain the main motivations and contributions of this paper. See the highlighted part of revised version. 

 Some comparisons should be given to show the advantage of this paper.

Reply to the Reviewer : we have added new paragraph in the revised version which clearly explain the advantage of this paper of this paper. See the highlighted part of revised version. 

 Do you provide some examples to show the effectiveness of main results?

Reply to the Reviewer: Respected professor, we have added one theoretical example in the revised version. 

Reviewer≠3: Comments to the Author

The manuscript is well-structured and clear, with precise mathematical formulations and thorough proofs. The abstract, discussion, and conclusion effectively summarize the research. However, the authors should revise the introduction section to clearly state the motivation behind this new work and explicitly address the gap between existing research and the new contributions presented. Additionally, the introduction should provide a clearer context by referencing relevant and recent articles related to their work.

1. Applications of a q-Differential Operator to a Class of Harmonic Mappings Defined by q-Mittag–Leffler Functions, Symmetry, 14, 1905 (2022), doi.org/10.3390/sym14091905.

2. Faber polynomial coefficients estimates for certain subclasses of q-Mittag-Leffler-Type analytic and bi-univalent functions, AIMS Mathematics, 7(2) (2021), 2512-2528.

3. Some Applications of Analytic Functions Associated with q-Fractional Operator, Mathematics, 11, (2023), 930.https://doi.org/10.3390/ math110409.

Reply to the Reviewer: Thanks for highlighting the related article. We added these articles in our manuscript. Researchers can extend this work by using the concepts of above mentions papers. 

I would like to bring to your attention some errors and suggestions

1. In the abstract line 4: It should be “four-parameter Mittag-Leffler function ”

2. In the abstract, line 7: Replace “specified” with “proved”.

3. On page 9, line 23: Replace “specific instances “ with special cases”

4. On page 12, in the start of the line 5, remove the extra words “some new results”.

5. Check for typos error in the whole paper.

Reply to the Reviewer: We have incorporate the point 1-5, in the revised version.

6. The conclusion section lacks clarity on future directions. Please revise and expand the conclusions to include specific future research directions.

Decision: This paper by Abubaker et al. makes a substantial contribution to fractional calculus and Geometric Function Theory through differential subordination. The results are well formulated, new and easily understandable. After the above mentioned changes, the article can be published in “Plos One”.

Reply to the Reviewer: According to your suggestions, we have added future directions in the revised version, which are highlighted in revised version.

Reviewer≠4: Comments to the Author

The paper addresses a topic that provides very interesting results and is intensely investigated nowadays which concerns the embedding of fractional calculus aspects into geometric function theory. The introduction of new fractional operators is a research theme of interest at the moment and this is well evidenced in the text of the paper. The new integral operator introduced in this paper is valuable due to the properties proved in this paper which can open ways for future investigations by applying this operator in the theories of differential subordination and superordination and other fields that apply fractional operators. The motivation of the paper is clearly presented, all the necessary notions and previously established results are highlighted in the Introduction and all the references cited in the text are relevant. The paper is well organized and easy to follow. The proofs are correct and contain all the necessary details.

 The paper needs small adjustments regarding the English language. Most important, the Abstract should be rephrased.

 page 3: "Let H(U) represents" should be "Let H(U) represent" and the sentence should begin at a new paragraph. "...is the subclass of H(U) which is define as:" should be "...is the subclass of H(U) which is defined as:"

Reply to the Reviewer:

Respected Professor,

Thank you for your careful reading and remarks. We have made every effort to eliminate all typographical and language errors. Further, we have rephrased abstract according to your suggestion. All changes are highlighted in the revised version PDF.

---

## [Decision Letter · Decision Letter 1]

5 Jan 2025

Some new applications of the fractional integral and four-parameter Mittag-Leffler function

PONE-D-24-38839R1

Dear Dr. B. Al-Shaikh,

We’re pleased to inform you that your manuscript has been judged scientifically suitable for publication and will be formally accepted for publication once it meets all outstanding technical requirements.

Kind regards,

Behzad Ghanbari

Academic Editor

PLOS ONE

Additional Editor Comments (optional):

Reviewers' comments:

Reviewer's Responses to Questions

**Comments to the Author**

1. If the authors have adequately addressed your comments raised in a previous round of review and you feel that this manuscript is now acceptable for publication, you may indicate that here to bypass the “Comments to the Author” section, enter your conflict of interest statement in the “Confidential to Editor” section, and submit your "Accept" recommendation.

Reviewer #1: All comments have been addressed

Reviewer #3: All comments have been addressed

Reviewer #4: All comments have been addressed

2. Is the manuscript technically sound, and do the data support the conclusions?

Reviewer #1: Yes

Reviewer #3: Yes

Reviewer #4: Yes

3. Has the statistical analysis been performed appropriately and rigorously? 

Reviewer #1: N/A

Reviewer #3: N/A

Reviewer #4: N/A

4. Have the authors made all data underlying the findings in their manuscript fully available?

Reviewer #1: Yes

Reviewer #3: Yes

Reviewer #4: Yes

5. Is the manuscript presented in an intelligible fashion and written in standard English?

Reviewer #1: Yes

Reviewer #3: Yes

Reviewer #4: Yes

6. Review Comments to the Author

Reviewer #1: Taking account that the authors made the revision indicated by the reviewers, I recommend the paper for publication.

Reviewer #3: All changes have been incorporated into the revised version by the authors. Therefore at this stage no further revisions are required.

Reviewer #4: The authors have addressed all the reviewers' comments. The present form of the paper can be accepted for publication.

7. PLOS authors have the option to publish the peer review history of their article (what does this mean?). If published, this will include your full peer review and any attached files.

Reviewer #1: No

Reviewer #3: **Yes: **Dr. Nazar Khan

Reviewer #4: No

---

## [Editor Report · Acceptance letter]

23 Jan 2025

PONE-D-24-38839R1 

PLOS ONE

Dear Dr. B. Al-Shaikh, 

I'm pleased to inform you that your manuscript has been deemed suitable for publication in PLOS ONE. Congratulations! Your manuscript is now being handed over to our production team.

Kind regards, 

on behalf of

Professor Behzad Ghanbari 

Academic Editor

PLOS ONE